# ANOMALY DETECTION WITH GENERATIVE ADVERSARIAL NETWORKS

## ABSTRACT

Many anomaly detection methods exist that perform well on low-dimensional problems however there is a notable lack of effective methods for high-dimensional spaces, such as images. Inspired by recent successes in deep learning we propose a novel approach to anomaly detection using generative adversarial networks. Given a sample under consideration, our method is based on searching for a good representation of that sample in the latent space of the generator; if such a representation is not found, the sample is deemed anomalous. We achieve state-of-the-art performance on standard image benchmark datasets and visual inspection of the most anomalous samples reveals that our method does indeed return anomalies.

## 1 INTRODUCTION

Given a collection of data it is often desirable to automatically determine which instances of it are unusual. Commonly referred to as *anomaly detection*, this is a fundamental machine learning task with numerous applications in fields such as astronomy (Protopapas et al., 2006; Dutta et al., 2007), medicine (Campbell and Bennett, 2001; Wong et al., 2003; Schlegl et al., 2017), fault detection (Görnitz et al., 2015), and intrusion detection (Eskin, 2000; Hu et al., 2003). Traditional algorithms often focus on the low-dimensional regime and face difficulties when applied to high-dimensional data such as images or speech. Second to that, they require the manual engineering of features.

Deep learning omits manual feature engineering and has become the de-facto approach for tackling many high-dimensional machine learning tasks. The latter is largely a testament of its experimental performance: deep learning has helped to achieve impressive results in image classification (Krizhevsky et al., 2012), and is setting new standards in domains such as natural language processing (Le and Mikolov, 2014; Sutskever et al., 2014) and speech recognition (Bahdanau et al., 2015).

In this paper we present a novel deep learning based approach to anomaly detection which uses generative adversarial networks (GANs) (Goodfellow et al., 2014). GANs have achieved state-of-the-art performance in high-dimensional generative modeling. In a GAN, two neural networks – the discriminator and the generator – are pitted against each other. In the process the generator learns to map random samples from a low-dimensional to a high-dimensional space, mimicking the target dataset. If the generator has successfully learned a good approximation of the training data's distribution it is reasonable to assume that, for a sample drawn from the data distribution, there exists some point in the GAN's latent space which, after passing it through the generator network, should closely resembles this sample. We use this correspondence to perform anomaly detection with GANs (ADGAN).

In Section 2 we give an overview of previous work on anomaly detection and discuss the modeling assumptions of this paper. Section 3 contains a description of our proposed algorithm. In our experiments, see Section 4, we both validate our method against traditional methods and showcase ADGAN's ability to detect anomalies in high-dimensional data.

## 2 BACKGROUND

Here we briefly review previous work on anomaly detection, touch on generative models, and highlight the methodology of GANs.

### 2.1 RELATED WORK

**Anomaly detection.** Research on anomaly detection has a long history with early work going back as far as Edgeworth (1887), and is concerned with finding unusual or *anomalous* samples in a corpus of data. An extensive overview over traditional anomaly detection methods as well as open challenges can be found in Chandola et al. (2009). For a recent empirical comparison of various existing approaches, see Emmott et al. (2013).

Generative models yield a whole family of anomaly detectors through estimation of the data distribution $p$. Given data, we estimate $\hat{p} \approx p$ and declare those samples which are unlikely under $\hat{p}$ to be anomalous. This guideline is roughly followed by traditional non-parametric methods such as kernel density estimation (KDE) (Parzen, 1962), which were applied to intrusion detection in Yeung and Chow (2002). Other research targeted mixtures of Gaussians for active learning of anomalies (Pelleg and Moore, 2005), hidden Markov models for registering network attacks (Ourston et al., 2003), and dynamic Bayesian networks for traffic incident detection (Singliar and Hauskrecht, 2006).

**Deep generative models.** Recently, variational autoencoders (VAEs) (Kingma and Welling, 2013) have been proposed as a deep generative model. By optimizing over a variational lower bound on the likelihood of the data, the parameters of a neural network are tuned in such a way that samples resembling the data may be generated from a Gaussian prior. Another generative approach is to train a pair of deep convolutional neural networks in an autoencoder setup (DCAE) (Masci et al., 2011) and producing samples by decoding random points on the compression manifold. Unfortunately, none of these approaches yield a tractable way of estimating $p$. Our approach uses a deep generative model in the context of anomaly detection.

**Deep learning for anomaly detection.** Non-parametric anomaly detection methods suffer from the curse of dimensionality and are thus inadequate tools for the interpretation and analysis of high-dimensional data. Deep neural networks have been found to obviate many problems that arise in this context. As a hybrid between the two approaches, deep belief networks were coupled with one-class support vector machines to detect anomalies in Erfani et al. (2016). We found that this technique did not work well for image datasets, and indeed the authors included no such experiments in their paper. Similarly, one may employ a network that was pretrained on a different task (such as classification on ImageNet) and then use this network's intermediate features to extract relevant information from images. We tested this an approach in our experimental section.

Recently GANs, which we discuss in greater depth in the next section, have garnered much attention with performance surpassing previous deep generative methods. Concurrently to this work, Schlegl et al. (2017) developed an anomaly detection framework that uses GANs in a similar way as we do. We discuss the differences between our work and theirs in Section 3.2.

### 2.2 GENERATIVE ADVERSARIAL NETWORKS

GANs, which lie at the heart of ADGAN, have set a new state-of-the-art in generative image modeling. They provide a framework to generate samples that are approximately distributed to $p$, the distribution of the training data $\{x_i\}_{i=1}^n \triangleq \mathcal{X} \subseteq \mathbb{R}^d$. To achieve this, GANs attempt to learn the parametrization of a neural network, the so-called generator $g_\theta$, that maps low-dimensional samples drawn from some simple noise prior $p_z$ (e.g. a multivariate Gaussian) to samples in the image space, thereby inducing a distribution $q_\theta$ (the push-forward of $p_z$ with respect to $g_\theta$) that approximates $p$. To achieve this a second neural network, the discriminator $d_\omega$, learns to classify the data from $p$ and $q_\theta$. Through an alternating training procedure the discriminator becomes better at separating

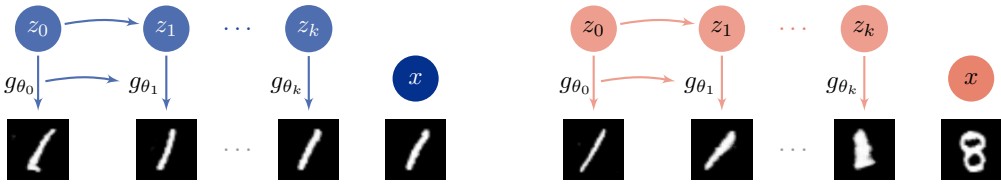

Figure 1: An illustration of ADGAN. In this example, ones from MNIST are considered normal. After an initial draw from $p_z$, the loss between the first generation $g_{\theta_0}(z_0)$ and the image $x$ whose anomaly we are assessing is computed. This information is used to generate a consecutive image $g_{\theta_1}(z_1)$ more alike $x$. After $k$ steps, samples are scored. If $x$ is similar to the training data (blue example), then a similar object should be contained in the image of $g_{\theta_k}$. For a dissimilar $x$ (red example), no similar image is found, resulting in a large loss.

samples from $p$ and samples from $q_\theta$, while the generator adjusts $\theta$ to fool the discriminator, thereby approximating $p$ more closely. The objective function of the GAN framework is thus:

$$\min_\theta \max_\omega \left\{ V(\theta, \omega) = \mathbb{E}_{x \sim p}[\log d_\omega(x)] + \mathbb{E}_{z \sim p_z}[\log(1 - d_\omega(g_\theta(z)))] \right\}, \tag{1}$$

where $z$ are vectors that reside in a latent space of dimensionality $d' \ll d$.[1] A recent work showed that this minmax optimization (1) equates to an empirical lower bound of an $f$-divergence (Nowozin et al., 2016).[2]

GAN training is difficult in practice, which has been shown to be a consequence of vanishing gradients in high-dimensional spaces (Arjovsky and Bottou, 2017). These instabilities can be countered by training on integral probability metrics (IPMs) (Müller, 1997; Sriperumbudur et al., 2009), one instance of which is the 1-Wasserstein distance.[3] This distance, informally defined, is the amount of work to pull one density onto another, and is the basis of the Wasserstein GAN (WGAN) (Arjovsky et al., 2017). The objective function for WGANs is

$$\min_\theta \max_{\omega \in \Omega} \left\{ W(\theta, \omega) = \mathbb{E}_{x \sim p}[d_\omega(x)] - \mathbb{E}_{z \sim p_z}[d_\omega(g_\theta(z))] \right\}, \tag{2}$$

where the parametrization of the discriminator is restricted to allow only 1-Lipschitz functions, i.e. $\Omega = \{\omega \colon \|d_\omega\|_{\mathrm{L}} \le 1\}$. When compared to classic GANs, we have observed that WGAN training is extremely stable and is thus used in our experiments, see Section 4.

## 3 ALGORITHM

Our proposed method (ADGAN, see Alg. 1) sets in after GAN training has converged. If the generator has indeed captured the distribution of the training data then, given a new sample $x \sim p$, there should exist a point $z$ in the latent space, such that $g_\theta(z) \approx x$. Additionally we expect points away from the support of $p$ to have no representation in the latent space, or at least occupy a small portion of the probability mass in the latent distribution, since they are easily discerned by $d_\omega$ as not coming from $p$. Thus, given a test sample $x$, if there exists no $z$ such that $g_\theta(z) \approx x$, or if such a $z$ is difficult to find, then it can be inferred that $x$ is not distributed according to $p$, i.e. it is anomalous. Our algorithm hinges on this hypothesis, which we illustrate in Fig. 1.

---

[1]That $p$ may be approximated via transformations from a low-dimensional space is an assumption that is implicitly motivated from the manifold hypothesis (Narayanan and Mitter, 2010).

[2]This lower bound becomes tight for an optimal discriminator, making apparent that $V(\theta, \omega^*) \propto \mathrm{JS}[p|q_\theta]$.

[3]This is achieved by restricting the class over which the IPM is optimized to functions that have Lipschitz constant less than one. Note that in Wasserstein GANs, an expression corresponding to a lower bound is optimized instead.

---

**Algorithm 1:** Anomaly Detection using Generative Adversarial Networks (ADGAN).

**Input**: parameters $(\gamma, \gamma_\theta, n_{\text{seed}}, k)$, sample $x$, GAN generator $g_\theta$, prior $p_z$, reconstruction loss $\ell$.

1   initialize $\{z_{j,0}\}_{j=1}^{n_{\text{seed}}} \sim p_z$ and $\{\theta_{j,0}\}_{j=1}^{n_{\text{seed}}} \triangleq \theta$.
2   **for** $j = 1, \ldots, n_{\text{seed}}$ **do**
3     **for** $l = 1, \ldots, k$ **do**
4       $z_{j,l} \leftarrow z_{j,l-1} - \gamma \cdot \nabla_{z_{j,l-1}} \ell(g_{\theta_{j,l-1}}(z_{j,l-1}), x)$
5       $\theta_{j,l} \leftarrow \theta_{j,l-1} - \gamma_\theta \cdot \nabla_{\theta_{j,l-1}} \ell(g_{\theta_{j,l-1}}(z_{j,l-1}), x)$
6     **end**
7   **end**
8   return $(1/n_{\text{seed}}) \sum_{j=1}^{n_{\text{seed}}} \ell(g_{\theta_{j,k}}(z_{j,k}), x)$.

---

### 3.1 ADGAN

To find $z$, we initialize from $z_0 \sim p_z$, where $p_z$ is the same noise prior also used during GAN training. For $l = 1, \ldots, k$ steps, we backpropagate the reconstruction loss $\ell$ between $g_\theta(z_l)$ and $x$, making the subsequent generation $g_\theta(z_{l+1})$ more like $x$. At each iteration, we also allow a small amount of flexibility to the parametrization of the generator, resulting in a series of mappings from the latent space $g_{\theta_0}(z_0), \ldots, g_{\theta_k}(z_k)$ that more and more closely resembles $x$. Adjusting $\theta$ gives the generator additional representative capacity, which we found to improve the algorithm's performance. Note that these adjustments to $\theta$ are not part of the GAN training procedure and $\theta$ is reset back to its original trained value for each new testing point.

To limit the risk of seeding in unsuitable regions and address the non-convex nature of the underlying optimization problem, the search is initialized from $n_{\text{seed}}$ individual points. The key idea underlying ADGAN is that if the generator was trained on the same distribution $x$ was drawn from, then the average over the final set of reconstruction losses $\{\ell(x, g_{\theta_{j,k}}(z_{j,k}))\}_{j=1}^{n_{\text{seed}}}$ will assume low values, and high values otherwise.

Our method may also be understood from the standpoint of approximate inversion of the generator. In this sense, the above backpropagation finds latent vectors $z$ that lie close to $g_\theta^{-1}(x)$. Inversion of the generator was previously studied in Creswell and Bharath (2016), where it was verified experimentally that this task can be carried out with high fidelity. In addition Lipton and Tripathi (2017) showed that generated images can be successfully recovered by backpropagating through the latent space.[4] Jointly optimizing latent vectors and the generator parametrization via backpropagation of reconstruction losses was investigated in detail by Bojanowski et al. (2017). The authors found that it is possible to train the generator entirely without a discriminator, still yielding a model that incorporates many of the desirable properties of GANs, such as smooth interpolations between samples.

### 3.2 ALTERNATIVE APPROACHES

Given that GAN training also gives us a discriminator for discerning between real and fake samples, one might reasonably consider directly applying the discriminator for detecting anomalies. However, once converged, the discriminator exploits checkerboard-like artifacts on the pixel level, induced by the generator architecture (Odena et al., 2016; Lopez-Paz and Oquab, 2017). While it perfectly separates real from forged data, it is not equipped to deal with samples which are completely unlike the training data. This line of reasoning is verified in Section 4 experimentally.

Another approach we considered was to evaluate the likelihood of the final latent vectors $\{z_{j,k}\}_{j=1}^{n_{\text{seed}}}$ under the noise prior $p_z$. This approach was tested experimentally in Section 4, and while it showed some promise, it was consistently outperformed by ADGAN.

In Schlegl et al. (2017), the authors propose a technique for anomaly detection (called AnoGAN) which uses GANs in a way somewhat similar to our proposed algorithm. Their algorithm also begins

---

[4]While it was shown that any $g_\theta(z)$ may be reconstructed from some other $z_0 \in \mathbb{R}^{d'}$, this does not mean that the same holds for an $x$ not in the image of $g_\theta$.

by training a GAN. In a manner similar to our own, given a test point $x$, their algorithm searches for a point $z$ in the latent space such that $g_\theta(z) \approx x$ and computes the reconstruction loss. Additionally they use an intermediate discriminator layer $d'_\omega$ and compute the loss between $d'_\omega(g_\theta(z))$ and $d'_\omega(x)$. They use a convex combination of these two quantities as their anomaly score.

In ADGAN we never use the discriminator, which is discarded after training. This makes it easy to couple ADGAN with any GAN-based approach, e.g. LSGAN (Mao et al., 2016), but also any other differentiable generator network such as VAEs or moment matching networks (Li et al., 2015). In addition, we account for the non-convexity of the underlying optimization by seeding from multiple areas in the latent space. Lastly, during inference we update not only the latent vectors $z$, but jointly update the parametrization $\theta$ of the generator.

## 4   EXPERIMENTS

Here we present experimental evidence of the efficacy of ADGAN. We compare our algorithm to competing methods on a controlled, classification-type task and show anomalous samples from popular image datasets. Our main findings are that ADGAN:

- outperforms non-parametric as well as available deep learning approaches on two controlled experiments where ground truth information is available;
- may be used on large, unsupervised data (such as LSUN bedrooms) to detect anomalous samples that coincide with what we as humans would deem unusual.

### 4.1   DATASETS

Our experiments are carried out on three benchmark datasets with varying complexity: (i.) MNIST (LeCun, 1998) which contains grayscale scans of handwritten digits. (ii.) CIFAR-10 (Krizhevsky and Hinton, 2009) which contains color images of real world objects belonging to ten classes. (iii.) LSUN (Xiao et al., 2010), a dataset of images that show different scenes (such as bedrooms, bridges, or conference rooms). For all datasets the training and test splits remain as their default. In addition, all images are rescaled to assume pixel values in $[-1, 1]$.

### 4.2   METHODS AND HYPERPARAMETERS

We tested the performance of ADGAN against three traditional, non-parametric approaches commonly used for anomaly detection: (i.) KDE with a Gaussian kernel (Parzen, 1962). The bandwidth is determined from maximum likelihood estimation over ten-fold cross validation, with $h \in \{2^0, 2^{1/2}, \dots, 2^4\}$. (ii.) One-class support vector machine (OC-SVM) (Schölkopf et al., 1999) with a Gaussian kernel. The inverse length scale is selected from estimating performance on a small holdout set of 1000 samples, and $\gamma \in \{2^{-7}, 2^{-6}, \dots, 2^{-1}\}$. (iii.) Isolation forest (IF), which was largely stable to changes in its parametrization. (iv.) Gaussian mixture model (GMM). We allowed the number of components to vary over $\{2, 3, \dots, 20\}$ and selected suitable hyperparameters by evaluating the Bayesian information criterion.

For the methods above we reduced the feature dimensionality before performing anomaly detection. This was done via PCA (Pearson, 1901), varying the dimensionality over $\{20, 40, \dots, 100\}$; we simply report the results for which best performance on a small holdout set was attained. As an alternative to a linear projection, we evaluated the performance of both methods after applying a non-linear transformation to the image data instead via an Alexnet (Krizhevsky et al., 2012), pretrained on Imagenet (Deng et al., 2009). Just as on images, the anomaly detection is carried out on the representation in the final convolutional layer of Alexnet. This representation is then projected down via PCA, as otherwise the runtime of KDE and OC-SVM becomes problematic.

We also report the performance of two end-to-end deep learning approaches: VAEs and DCAEs. For the DCAE we scored according to reconstruction losses, interpreting a high loss as indicative of a new sample differing from samples seen during training. In VAEs we scored by evaluating the evidence lower bound (ELBO). We found this to perform much better than thresholding directly via the prior likelihood in latent space or other more exotic approaches, such as scoring from the variance of the inference network.

In both DCAEs and VAEs we use a convolutional architecture similar to that of DCGAN (Radford et al., 2015), with batch norm regularizations (Ioffe and Szegedy, 2015) and ReLU activations in each layer. We also report the performance of AnoGAN. To put it on equal footing, we pair it with DCGAN (Radford et al., 2015), the same architecture also used for training in our approach.

ADGAN requires a trained generator. For this purpose, we trained on the WGAN objective (2), as this was much more stable than using GANs. The architecture was fixed to that of DCGAN (Radford et al., 2015). Following Metz et al. (2016) we set the dimensionality of the latent space to $d' = 256$.

For ADGAN, the searches in the latent space were initialized from the same noise prior that the GAN was trained on (in our case a normal distribution). To take into account the non-convexity of the problem, we seeded from $n_{\text{seed}} = 8$ points. For the optimization of latent vectors and the parameters of the generator we used the Adam optimizer (Kingma and Ba, 2014).[5] When searching for a point in in the latent space to match a test point, we found that more optimization steps always improved the performance in our experiments. We found $k = 5$ steps to be a good trade-off between execution time and accuracy and used this value in the results we report. Unless otherwise noted, we measured reconstruction quality with a squared $L_2$ loss.

## 4.3 ONE-VERSUS-ALL CLASSIFICATION

The first task is designed to quantify the performance of competing methods. In it, we closely follow the original publication on OC-SVMs (Schölkopf et al., 1999) and begin by training each model on data from a single class from MNIST. We then evaluate performance on 5000 items randomly selected from the test set, which contains samples from all classes. In each trial, we label the classes unseen in training as anomalous.

Ideally, a method assigns images from anomalous classes (say, digits 1-9) a higher anomaly score than images belonging to the normal class (zeros). Varying the decision threshold yields the receiver operating characteristic (ROC), shown in Fig. 2. In Table 1 and 2, we report the AUCs that resulted from leaving out each class. The second experiment follows this guideline with the colored images from CIFAR-10.

In these controlled experiments we highlight the ability of ADGAN to perform on-par with traditional methods at the task of inferring anomaly of low-dimensional samples such as those contained in MNIST. On CIFAR-10 we see that all tested methods see a drop in performance. For these experiments ADGAN performed best, needing eight seeds to achieve this result. Using a non-linear transformation with a pretrained Alexnet did not improve the performance of either MNIST or CIFAR10, see Table 1.

While neither table explicitly contains results from scoring the samples using the GAN discriminator, we did run these experiments for both datasets. Performance was weak, with an average AUC of 0.625 for MNIST and 0.513 for CIFAR-10. Scoring according to the prior likelihood $p_z$ of the final latent vectors worked slightly better, resulting in an average AUC of 0.721 for MNIST and 0.554 for CIFAR-10.

## 4.4 UNSUPERVISED ANOMALY DETECTION

In the second task we showcase the use of ADGAN in a practical setting where no ground truth information is available. For this we first trained a generator on LSUN scenes. We then used ADGAN to find the most anomalous images within the corresponding validation sets containing 300 images.[6] The images associated with the highest and lowest anomaly scores are shown in Fig. 3 and Fig. 4. It should be noted that the training set sizes studied in this experiment prohibit the use of non-parametric methods such as KDE and OC-SVMs.

---

[5]From a quick parameter sweep, we set the learning rate to $\gamma = 0.25$ and $(\beta_1, \beta_2) = (0.5, 0.999)$. We update the generator with $\gamma_\theta = 5 \cdot 10^{-5}$, the default learning rate recommended in Arjovsky et al. (2017).

[6]To quantify the performance on LSUN, we build a test set from combining the 300 validation samples of each scene. After training the generator on bedrooms only we recorded whether ADGAN assigns them low anomaly scores, while assigning high scores to samples showing any of the remaining scenes. This resulted in an AUC of 0.641.

| DATASET | $y_c$ | KDE | | OC-SVM | | IF | GMM |
|---|---|---|---|---|---|---|---|
| | | PCA | Alexnet | PCA | Alexnet | | |
| MNIST | 0 | 0.982 | 0.634 | **0.994** | 0.895 | 0.957 | 0.970 |
| | 1 | 0.999 | 0.922 | 0.999 | **1.000** | **1.000** | 0.999 |
| | 2 | 0.888 | 0.654 | **0.993** | 0.796 | 0.822 | 0.931 |
| | 3 | 0.898 | 0.639 | 0.933 | 0.932 | 0.924 | **0.951** |
| | 4 | 0.943 | 0.676 | 0.960 | 0.950 | 0.922 | **0.968** |
| | 5 | **0.930** | 0.651 | 0.898 | 0.855 | 0.859 | 0.917 |
| | 6 | 0.972 | 0.636 | **0.998** | 0.971 | 0.903 | 0.994 |
| | 7 | 0.933 | 0.628 | **0.946** | 0.884 | 0.938 | 0.938 |
| | 8 | **0.924** | 0.617 | 0.898 | 0.751 | 0.814 | 0.889 |
| | 9 | 0.940 | 0.644 | 0.942 | 0.959 | 0.913 | **0.962** |
| | | 0.941 | 0.670 | 0.945 | 0.899 | 0.905 | **0.952** |
| CIFAR-10 | 0 | 0.705 | 0.559 | 0.666 | 0.594 | 0.630 | **0.709** |
| | 1 | 0.493 | 0.487 | 0.473 | **0.540** | 0.379 | 0.443 |
| | 2 | **0.734** | 0.582 | 0.675 | 0.588 | 0.630 | 0.697 |
| | 3 | 0.522 | 0.531 | 0.530 | **0.575** | 0.408 | 0.445 |
| | 4 | 0.691 | 0.651 | **0.827** | 0.753 | 0.764 | 0.761 |
| | 5 | 0.439 | 0.551 | 0.438 | **0.558** | 0.514 | 0.505 |
| | 6 | 0.771 | 0.613 | **0.787** | 0.692 | 0.666 | 0.766 |
| | 7 | 0.458 | **0.593** | 0.532 | 0.547 | 0.480 | 0.496 |
| | 8 | 0.595 | 0.600 | **0.720** | 0.630 | 0.651 | 0.646 |
| | 9 | 0.490 | 0.529 | 0.453 | **0.530** | 0.459 | 0.384 |
| | | 0.590 | 0.570 | **0.610** | 0.601 | 0.558 | 0.585 |

Table 1: ROC-AUC of classic anomaly detection methods. For both MNIST and CIFAR-10, each model was trained on every class, as indicated by $y_c$, and then used to score against remaining classes.

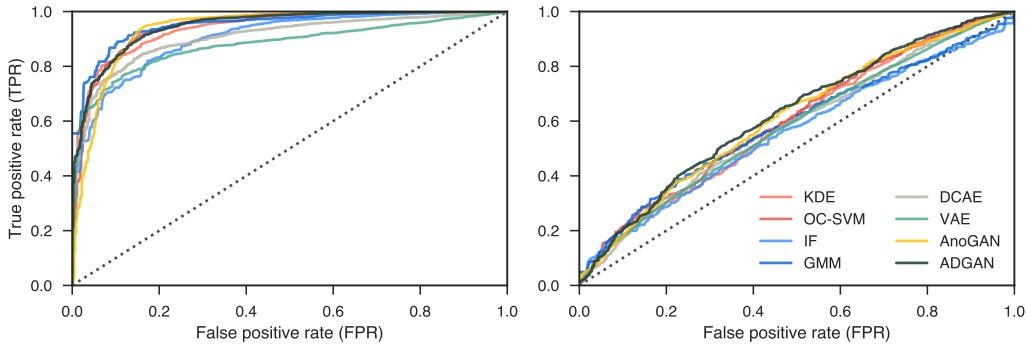

Figure 2: ROC curves for one-versus-all prediction of competing methods, averaged over all classes. The figure on the left contains results for MNIST, that on the right those for CIFAR-10. ADGAN is shown with $n_{\text{seed}} = 8$. KDE and OC-SVM are shown in conjunction with PCA.

As can be seen from visually inspecting the LSUN scenes flagged as anomalous, our method has the ability to discern usual from unusual samples. We infer that ADGAN is able to incorporate many properties of an image. It does not merely look at colors, but also takes into account whether shown geometries are canonical, or whether an image contains a foreign object (like a caption). Opposed to this, samples that are assigned a low anomaly score are in line with a classes' *Ideal Form*. They show plain colors, are devoid of foreign objects, and were shot from conventional angles. In the case of bedrooms, some of the least anomalous samples are literally just a bed in a room.

| DATASET | $y_c$ | DCAE | VAE | AnoGAN | ADGAN ($n_{seed} = 1$) | ADGAN ($n_{seed} = 8$) |
|---|---|---|---|---|---|---|
| MNIST | 0 | 0.988 | 0.921 | 0.990 | 0.972 | **0.995** |
| | 1 | 0.993 | **0.999** | 0.998 | 0.997 | **0.999** |
| | 2 | 0.917 | 0.815 | 0.888 | 0.874 | **0.936** |
| | 3 | 0.885 | 0.814 | 0.913 | 0.848 | **0.921** |
| | 4 | 0.862 | 0.879 | **0.944** | 0.910 | 0.936 |
| | 5 | 0.858 | 0.811 | 0.912 | 0.916 | **0.944** |
| | 6 | 0.954 | 0.943 | 0.925 | 0.957 | **0.967** |
| | 7 | 0.940 | 0.886 | 0.964 | 0.937 | **0.968** |
| | 8 | 0.823 | 0.780 | **0.883** | 0.816 | 0.854 |
| | 9 | **0.965** | 0.920 | 0.958 | 0.924 | 0.957 |
| | | 0.919 | 0.877 | 0.937 | 0.915 | **0.947** |
| CIFAR-10 | 0 | **0.656** | 0.620 | 0.610 | 0.627 | 0.632 |
| | 1 | 0.435 | **0.664** | 0.565 | 0.546 | 0.529 |
| | 2 | 0.381 | 0.382 | **0.648** | 0.561 | 0.580 |
| | 3 | 0.545 | 0.586 | 0.528 | 0.595 | **0.606** |
| | 4 | 0.288 | 0.386 | **0.670** | 0.586 | 0.607 |
| | 5 | 0.643 | 0.586 | 0.592 | 0.628 | **0.659** |
| | 6 | 0.509 | 0.565 | **0.625** | 0.604 | 0.611 |
| | 7 | **0.690** | 0.622 | 0.576 | 0.623 | 0.630 |
| | 8 | 0.698 | 0.663 | 0.723 | 0.702 | **0.744** |
| | 9 | 0.705 | **0.737** | 0.582 | 0.591 | 0.644 |
| | | 0.583 | 0.581 | 0.612 | 0.606 | **0.624** |

Table 2: ROC-AUC of deep anomaly detection methods.

Additional images that were retrieved from applying our method to CIFAR-10 and additional LSUN scenes have been collected into the Appendix.

## 5 CONCLUSION

We showed that searching the latent space of the generator can be leveraged for use in anomaly detection tasks. To that end, our proposed method: (i.) delivers state-of-the-art performance on standard image benchmark datasets; (ii.) can be used to scan large collections of unlabeled images for anomalous samples.

To the best of our knowledge we also reported the first results of using VAEs for anomaly detection. We remain optimistic that boosting its performance is possible by additional tuning of the underlying neural network architecture or an informed substitution of the latent prior.

Accounting for unsuitable initializations by jointly optimizing latent vectors and generator parameterization are key ingredients to help ADGAN achieve strong experimental performance. Nonetheless, we are confident that approaches such as initializing from an approximate inversion of the generator as in ALI (Donahue et al., 2016; Dumoulin et al., 2016), or substituting the reconstruction loss for a more elaborate variant, such as the Laplacian pyramid loss (Ling and Okada, 2006), can be used to improve our method further.

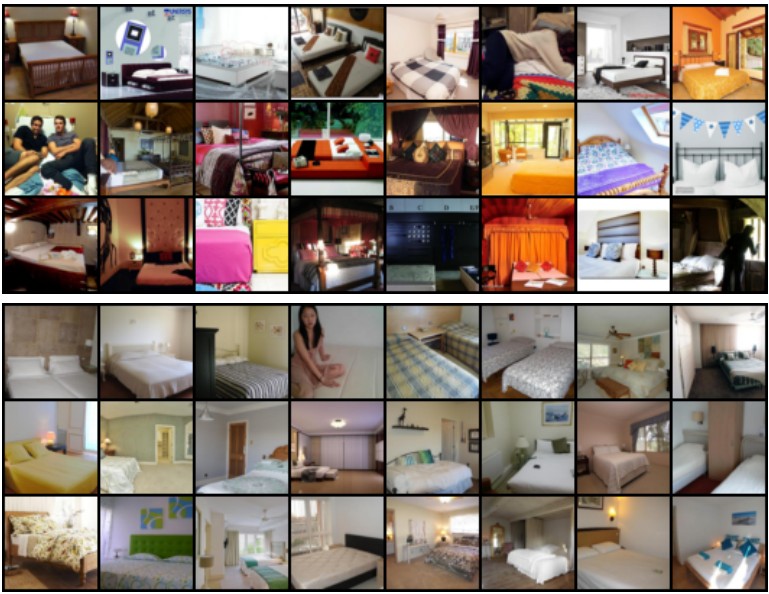

Figure 3: Starting from the top left, the first three rows show samples contained in the LSUN bedrooms validation set which, according to ADGAN, are the most anomalous (have the highest anomaly score). Again starting from the top left corner, the bottom rows contain images deemed normal (have the lowest score).

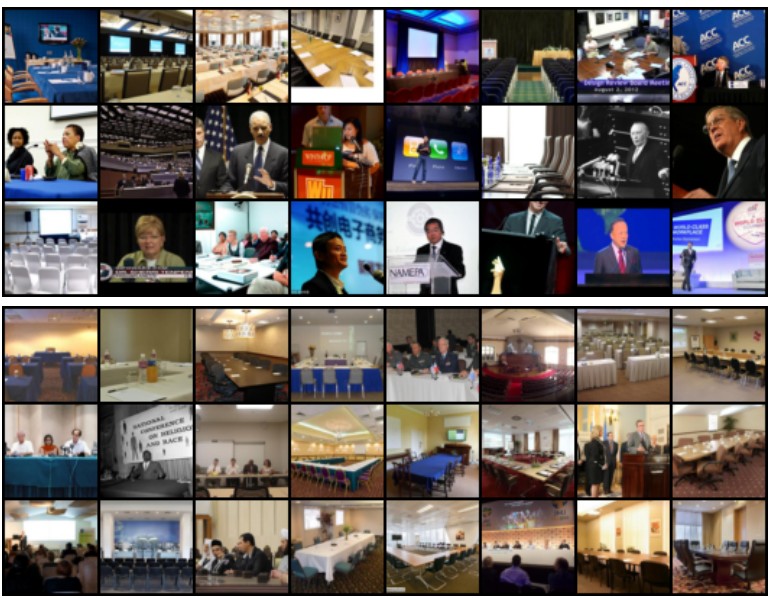

Figure 4: Scenes from LSUN showing conference rooms ranked by ADGAN. The top rows contain anomalous samples, the bottom rows scenes categorized as normal.

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

## A    ADDITIONAL LSUN SCENES

We ran ADGAN on additional scenes from the LSUN dataset showing bridges (Fig. 5), churches (Fig. 6), and restaurants (Fig. 7). Training was performed using DCGAN (Radford et al., 2015) on the WGAN objective (Arjovsky et al., 2017). Reconstruction losses were then measured on the 300 samples contained in the respective validation sets.

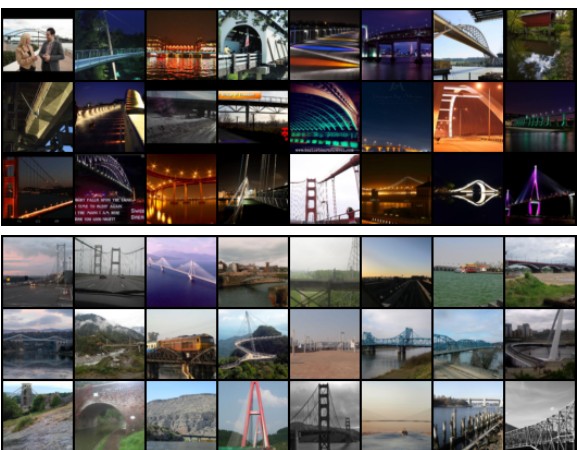

Figure 5: Additional scenes from LSUN that show images of bridges. The first rows show anomalous samples, the lower rows those deemed normal.

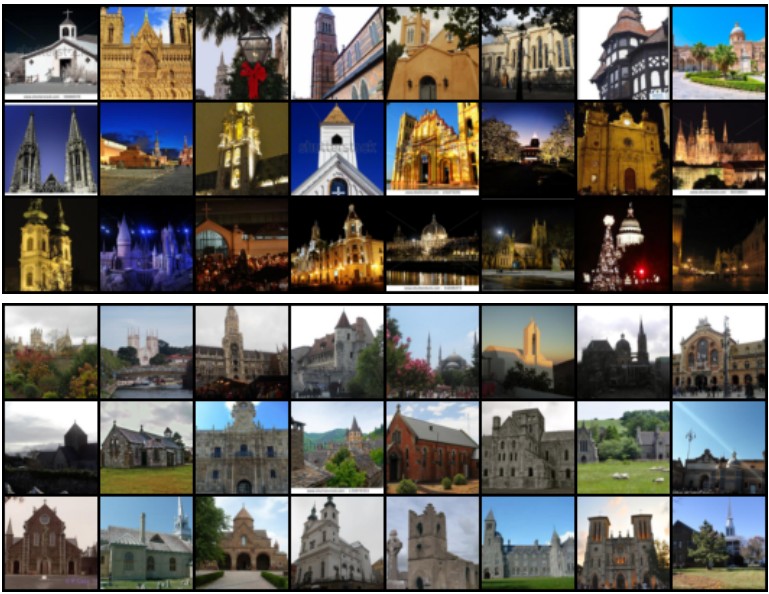

Figure 6: Images of churches, taken from LSUN. The first three rows show samples with high anomaly scores, the last three rows samples with a low score.

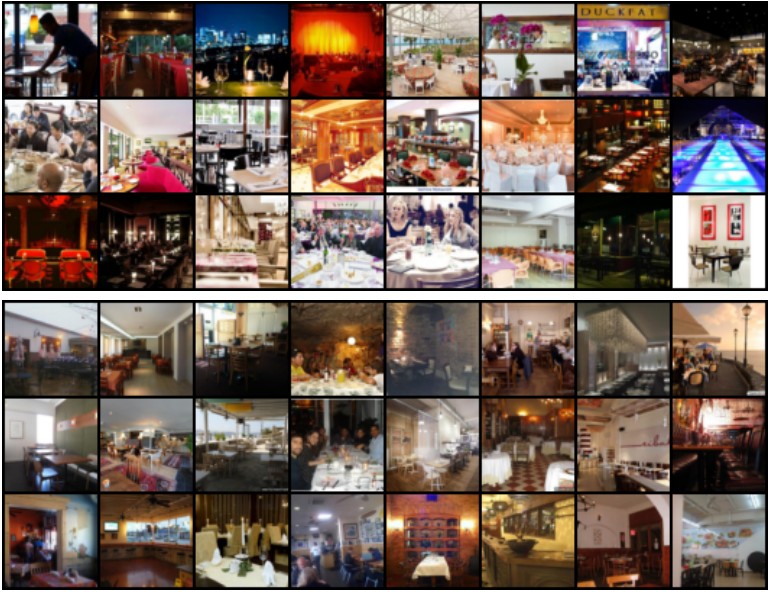

Figure 7: Scenes from LSUN that show restaurants. The first rows contain samples with high anomaly scores, the last rows those with low scores.

# B  UNSUPERVISED ANOMALY DETECTION ON CIFAR-10

Shown are additional experiments in which we determine anomalous samples of different classes (e.g. birds, cats, dogs) contained in CIFAR-10. ADGAN was applied exactly as described in Section 3, with the search carried out for $k = 100$ steps. In Fig. 8 we report the highest and lowest reconstruction losses of images that were randomly selected from the test set, conditioned on the respective classes.

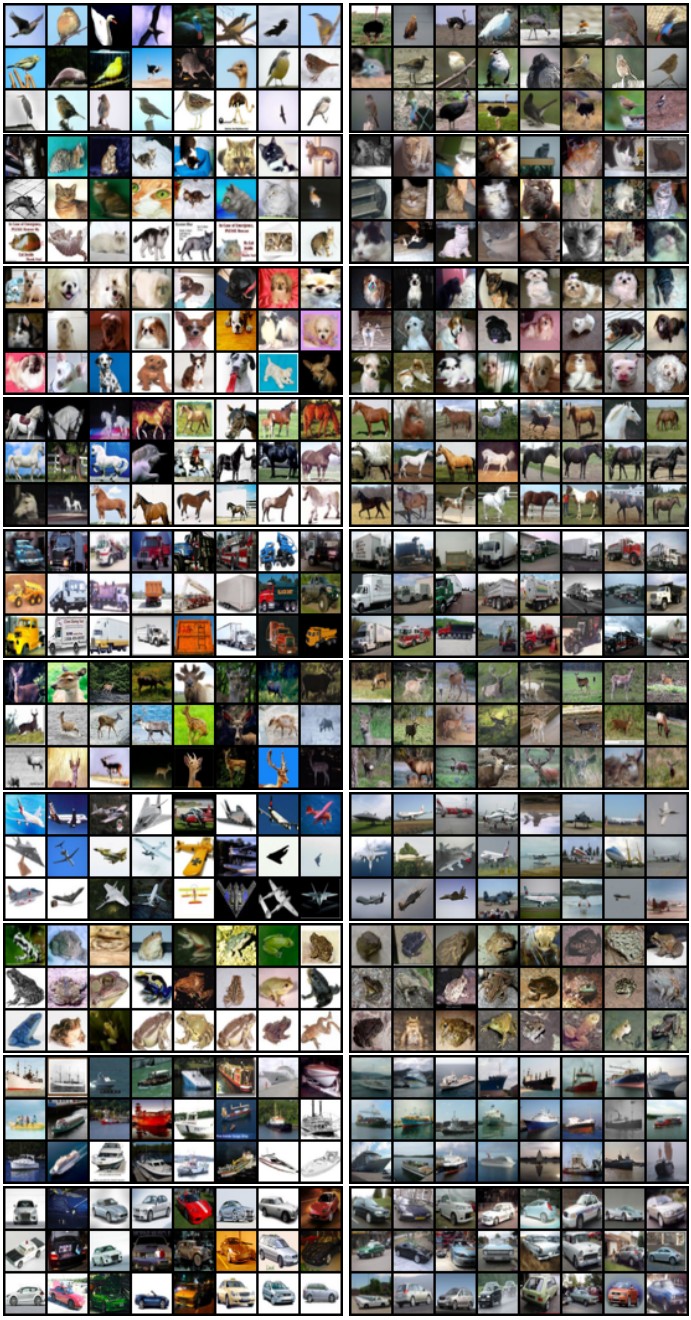

Figure 8: Each three rows, samples from a different class in the CIFAR-10 test set are shown. On the left, images deemed anomalous by ADGAN. The right column holds images considered canonical.

