# OpenReview forum: "Anomaly Detection with Generative Adversarial Networks"
_ICLR.cc/2018/Conference — Reject_

### Official Review · AnonReviewer2 · 2017-11-23
**Experimental baselines are too primitive.**

**Rating:** 4
**Confidence:** 5

**Review:**

In the paper, the authors proposed using GAN for anomaly detection.
In the method, we first train generator g_\theta from a dataset consisting of only healthy data points.
For evaluating whether the data point x is anomalous or not, we search for a latent representation z such that x \approx g_\theta(z).
If such a representation z could be found, x is deemed to be healthy, and anomalous otherwise.
For searching z, the authors proposed a gradient-descent based method that iteratively update z.
Moreover, the authors proposed updating the parameter \theta of the generator g_\theta.
The authors claimed that this parameter update is one of the novelty of their method, making it different from the method of Schlegl et al. (2017).
In the experiments, the authors showed that the proposed method attained the best AUC on MNIST and CIFAR-10.

In my first reading of the paper, I felt that the baselines in the experiments are too primitive.
Specifically, for KDE and OC-SVM, a naive PCA is used to reduce the data dimension.
Nowadays, there are several publicly available CNNs that are trained on large image datasets such as ImageNet.
Then, one can use such CNNs as feature extractor, that will give better low dimensional expression of the data than the naive PCA.
I believe that the performances of KDE and OC-SVM can be improved by using such feature extractors.

Additionally, I found that some well-known anomaly detection methods are excluded from the comparison.
In Emmott et al. (2013), which the authors referred as a related work, it was reported that Isolation Forest and Ensemble of GMMs performed well on several datasets (better than KDE and OC-SVM).
It would be essential to add these methods as baselines to be compared with the proposed method.

Overall, I think the experimental results are far from satisfactory.


### Response to Revision ###
It is interesting to see that the features extracted from AlexNet are not helpful for anomaly detection.
It would be interesting to see whether features extracted from middle layers are helpful or they are still useless.
I greatly appreciate the authors for their extensive experiments as a response to my comments.
However, I have decided to keep my score unchanged, as the additional experiments have shown that the performance of the proposed method is not significantly better than the other methods.
In particular, in MNIST, GMM performed better.

---

> ### Author Response · Authors · 2018-01-05
> **Response to AnonReviewer2**
>
> - We thank the reviewer for the recommendations for additional experimental baselines. We have now included all of the reviewer’s recommended baseline methods, other than the ensemble-GMM, where we used a standard GMM instead (we are not investigating ensemble methods, as many of these techniques have ensemble variants). Our method still performs the best.

---

### Official Review · AnonReviewer3 · 2017-11-26
**interesting paper with minor weakness**

**Rating:** 6
**Confidence:** 4

**Review:**

The paper is about doing anomaly detection for image data. The authors use a GAN based approach where it is trained in a standard way. After training is completed, the generator's latent space is explored to find a representation for a test image. Both the noise variable and generator model is updated using back propagation to achieve this. The paper is original, well written, easy to follow and presented ideas are interesting.

Strengths:
- anomaly detection for images is a difficult problem and the paper uses current state of the art in generative modeling (GAN) to perform anomaly detection.
- experiments section includes non-parametric methods such as OC-SVM as well as deep learning methods including a recent GAN based approach and the results are promising.

Weaknesses:
 - It is not clear why updating the generator during the anomaly detection helps. On evaluating on a large set of anomalies, the generator may run a risk of losing its ability to generate the original data if it is adjusted too much to the anomalies. The latent space of the generator no longer is the same that was achieved by training on the original data. I don't see how this is not a problem in the presented approach.
- The experimental results on data with ground truth needs statistical significance tests to convince that the benefits are indeed significant.
- It is not clear how the value of "k" (number of updates to the noise variable) was chosen and how sensitive it is to the performance. By having a large value of k, the reconstruction loss for an anomalous image may reduce to fall into the nominal category. How is this avoided?

---

> ### Author Response · Authors · 2018-01-05
> **Response to AnonReviewer3**
>
> Thanks to the reviewer for their helpful suggestions. Some remarks:
> - We actually reset the theta values to the original trained values for each testing sample before again optimizing them, so the theta adjustments made at testing time are not retained. We have updated our submission to make this more clear (page 4, paragraph 1).
> - We have updated our submission to explain our choice of the value for k (page 6, paragraph 3). Larger k always yields better performance, we chose k=5 for a good performance/evaluation time trade-off. In practice not many optimization steps are necessary before the performance gained from adding additional steps becomes small.

---

### Official Review · AnonReviewer1 · 2017-11-27
**anomaly detection scheme using GANs**

**Rating:** 4
**Confidence:** 4

**Review:**

Authors propose an anomaly detection scheme using GANs. It relies on a realistic assumption: points that are badly represented in the latent space of the generator are likely to be anomalous. Experiments are given in a classification and unsupervised context.
In the introduction, authors state that traditional algorithms "often fail when applied to high dimensional objects". Such claim should be supported by strong references as oc-svm or k-pca based anomaly detection algorithms (see Hoffman 2007) perform well in this context.
OC-SVM is a well-known technique that gives similar performances: authors fail at convincing that there are advantages of using the proposed framework, which do not strongly differs from previously published AnoGAN.
The underlying assumption of the algorithm (points badly represented by GANs are likely to be anomalous) justifies the fact that anomalies should be detected by the algorithm (type-I error). What is the rationale behind the type-II error? Is it expected to be small as well? What happens with adversarial examples for instance?

---

> ### Author Response · Authors · 2018-01-05
> **Response to AnonReviewer1**
>
> We thank the reviewer for their comments and suggestions.
> - Some traditional methods do indeed work well for high-dimensional learning and we have updated our paper to reflect that. We feel that it is accepted by the ML community that deep methods work the best for many high-dimensional tasks and we simply wanted to highlight that point.
> - We did not specifically address the type-II rationale in our original draft. We now highlight why we do not expect the generator to come up with an adequate representation of anomalous classes (page 3, final paragraph), thus giving some rationale for why the anomaly detector should avoid type-II errors.
> - ADGAN improves the CIFAR-10 AUC by about .014 over OC-SVM which is about an 13% improvement over the .5 random guessing baseline. (.624-.5)/(.610-.5)=1.127. We feel that this is a noteworthy improvement.

---

### Decision · Program_Chairs · 2018-01-29
**ICLR 2018 Conference Acceptance Decision**

**Decision:**

Reject

**Comment:**

The authors propose to detect anomaly based on its representation quality in the latent space of the GAN trained on valid samples.

Reviewers agree that:
- The proposed solution lacks novelty and similar approaches have been tried before.
- The baselines presented in the paper are primitive and hence do not demonstrate the clear benefits over traditional approaches.